# Strongly Metal-Adhesive and Self-Healing Gelatin@Polydopamine-Based Hydrogels with Long-Term Antioxidant Activity

**DOI:** 10.3390/antiox12091764

**Published:** 2023-09-14

**Authors:** Jordana Hirtzel, Guillaume Leks, Julie Favre, Benoît Frisch, Isabelle Talon, Vincent Ball

**Affiliations:** 13Bio, Laboratoire de Conception et Application de Molécules Bioactives, UMR 7199, Faculté de Pharmacie, Université de Strasbourg, CNRS, 74 Route du Rhin, 67401 Illkirch Cedex, France; j.hirtzel@unistra.fr (J.H.); guillaume.leks@etu.unistra.fr (G.L.); j.favre@unistra.fr (J.F.); frisch@unistra.fr (B.F.); 2Faculté de Chirurgie Dentaire, Université de Strasbourg, 8 Rue Sainte Elizabeth, 67000 Strasbourg, France; 3Biomatériaux & Bioingénierie, UMR_S 1121, Université de Strasbourg, INSERM, 1 Rue Eugène Boeckel, 67000 Strasbourg, France; isabelle.talon@chru-strasbourg.fr; 4Service de Chirurgie Pédiatrique, Hôpitaux Universitaire de Strasbourg, 67000 Strasbourg, France

**Keywords:** polydopamine, composite hydrogels, adhesion, peeling rate dependence, antioxidant

## Abstract

Bioinspired adhesives have been increasingly developed, especially towards a biomedical application. Therefore, in this study, dopamine (DA) was oxidized into polydopamine (PDA) in a gelatin mixture via titration with NaIO_4_ as a strong oxidant to easily obtain an adhesive antioxidant and self-healing PDA–gelatin hydrogel. Rheology experiments show a stiffness in the order of kPa and a thermal resistance above 50 °C, much above the gel–sol transition temperature of pristine gelatin. After heating at 55 °C, the gel is self-healing. In addition, just after formulation, it shows strong peeling-rate-dependent adhesion to steel with a tensile work per unit area (*W*) of up to 100 ± 39 J/m^2^, which is 2.5 times higher than that of the same gel without PDA at a peeling rate of 1000 µm/s. The increase in *W* between peeling rates of 10 and 1000 µm/s was studied and interpreted in terms of the gels’ viscoelasticity. Moreover, this hydrogel offers significant antioxidant activity (measured by DPPH scavenging) that lasts with storage for at least over 15 days, this being then prolonged for 2 additional days, which seems particularly relevant considering the importance of reactive oxygen species (ROS) in wound healing. To summarize, PDA–gelatin gel is a promising strong and antioxidant adhesive.

## 1. Introduction

Inspired by mussel adhesives originating from the presence of L-Lysine and L-DOPA-rich proteins [1] (called *mefps*), many synthetic adhesives have been developed mostly for biomedical applications [2] in order to replace staples or sutures. All these adhesives rely on the peculiar and versatile chemistry of catecholamine and catechol groups [3,4,5]. Even if some attempts have been made to use peptides containing sequences reminiscent of mussel foot proteins (*mefps*) [6,7], such strategies appeared highly expensive. Hence, much more investigations have been conducted to graft catechol groups as side chains of natural [8,9,10,11,12] and synthetic polymers [9,13,14,15,16,17]. Such grafting approaches require chemical modification and purifications steps to remove the unbound catechol groups. Even if these grafting–purification steps were most of the time relatively easy, it was asked if the simple one-pot blending of the polymer of interest, carrying nucleophilic groups able to bind with oxidized catechols, and if the required catechol or catecholamine could not provide adhesive strength of the same order of magnitude. Some feasibility concepts of that idea have been recently provided [18,19,20,21]. The single blending of catechols or catecholamines with a polymer able to undergo gelation will produce strong adhesives in a one-pot manner and allow investigations of the influence of the catecholamine structure and concentration on the hydrogel’s adhesive properties.

The aim of the present work is to complement our previous investigations in order not only to improve the adhesion strength of gelatin hydrogels blended with dopamine and sodium periodate (to yield polydopamine in the gel) but also to understand the influence of the peeling rate on the adhesion strength, which is a typical signature of viscoelastic adhesives [22,23]. In addition, we will show the self-healing behavior of those porcine gelatin@dopamine-based hydrogels and their related prolonged (over days) antioxidant properties. The provision of such a prolonged effect, rather than a burst release of antioxidants, is of major importance for bio-applications of those materials but also for the protection against corrosion [24]. In the biomedical field, gels with antioxidant properties are active wound-healing materials because they are able to reduce the concentration of reactive oxygen species (ROS) in the wound region. ROS are produced by macrophages and neutrophils in the injured area and are aimed at killing bacteria. However, an excessive ROS concentration increases the risk of infection and strong damage to the non-injured tissue around the wound [25]. Hence, a controlled balance of ROS may be required by extracellular additives when the enzymes of the considered tissue, like superoxide dismutase and catalase, are not able to ensure the appropriate ROS concentration. Hydrogels containing up to 0.5% (*w/w*) of polydopamine (PDA) nanoparticles have been shown to scavenge both hydroxyl and superoxide radicals, to reduce the intracellular ROS concentration in human umbilical vein endothelial cells (HUVEC) and to allow strong wound healing in Sprague Dawley male mice [26]. In a preceding paper, it was shown that polypyrrole-grafted gelatin hydrogels, crosslinked with Fe^3+^ cations and incorporating silver decorated PDA nanoparticles, were antioxidant and antibacterial due to the presence of PDA and silver, respectively [27]. Hydrogels made by blending chitosan and oxidized β-glucan were shown to be antioxidant and resorbable and useful for diabetic wound repair [28]. Polydopamine–agarose hydrogels were stabilized via the addition of ε-polylysine to display a stable photothermal effect and satisfactory antimicrobial properties [29]. However, the prolonged antioxidant activity of the polydopamine-containing hydrogels, over several days, which may be required for efficient wound healing, was not investigated. It is the major aim of this investigation to test the assumption that PDA–gelatin hydrogels prepared in a one-pot manner are able to display long-term antioxidant properties, in addition to strong adhesion to stainless steel.

## 2. Materials and Methods

All the reactants were purchased and used without additional purification. Gelatin from bovine (ref: G6650) and porcine skin (ref: G1890) were purchased from Sigma Aldrich (St. Louis, MO, USA). The bovine gelatin was of type B with a bloom degree of 250, whereas the porcine gelatin was of type A with a bloom degree of 300. We compared both gelatin gels for their mechanical properties in order to select the most rigid and thermally resistant one for adhesion to stainless steel and antioxidant property evaluations.

### 2.1. Gel Formulation

Dopamine hydrochloride (DA) (prepared to reach a final concentration of 1.5 mg/mL) (Sigma Aldrich, ref: H8502), and bovine or porcine gelatin (to reach 10% *w/w*) were dissolved in sodium acetate buffer (Merck, Darmstadt, Germany, ref: 1.06268, 50 mM, adjusted to pH = 5). This pH value was chosen because the autoxidation of dopamine (due to dissolved O_2_) is extremely slow under these conditions, and hence an external oxidant has to be added to trigger the oxidation of dopamine into PDA [30]. The suspension was magnetically stirred and heated until the temperature reached 45–50 °C and the gelatin was completely dissolved. This mixture was then titrated with a concentrated solution of NaIO_4_ (Sigma Aldrich, ref: 311448) under strong stirring to reach a final NaIO_4_/DA ratio of 2. The whole experimental procedure is shown in Figure 1.

The addition of this strong oxidant induced an immediate color shift from beige to deep red in relation to the oxidation of dopamine [30]. The obtained sol was then immediately poured onto the support, left to crosslink through DA oxidation for 1 h at room temperature and rinsed with MiliQ water. The obtained gels will be referred to as PDA–gelatin gels in the following and their properties will be compared to those of gelatin gels that are unmodified but incorporate the same amount of NaIO_4_.

### 2.2. Dehydration of the Hydrogels

The gels were dehydrated to estimate their self-healing properties upon rehydration. PDA–gelatin gels (2 and 4 mL) were formulated following the protocol described above and poured into Petri dishes and polypropylene sample containers. After 1 h of gelation and water rinsing, three protocols were evaluated to evaluate the possibility to dehydrate these high-water-content materials by weighing the gels before and after dehydration. The gels were either

i.Frozen at −80 °C for 4 h and freeze-dried overnight using an Alpha 1-4LD plus (Christ) device at 0.024 mbar;ii.Immersed in 30 mL of absolute EtOH over a weekend;iii.Put in an oven at 37 °C for 24 h.

No systematic swelling investigation was undertaken in this work because we aimed to use these gels as adhesives for metallic materials and not for bio-applications.

### 2.3. Rheology Experiments

All rheological experiments were performed with a Kinexus Ultra rheometer (Malvern, Great Britain). The maximum force and temperature measurable with this device were 50 N and 50 °C, respectively.

The gelation kinetics, frequency sweep, and thermal stability experiments were executed one after the other with the same sample of gel (1.3 mL) being poured in between stainless-steel cone-plate geometries. First, the viscoelastic properties of the gels were measured for 1 h at 25 °C, at a constant frequency of 1 Hz, and at a constant shear strain of 1%. Then, for the frequency sweep experiments shear stress was applied with the same set of parameters except for frequency, which was changed from 10 to 0.01 Hz (10 measurements per frequency decade), the temperature being held constant at 25 °C. Finally, the sample was submitted to a temperature ramp from 25 to 50 °C at 1 °C/min, the frequency and shear strain being constant and equal to 1 Hz and 1%, respectively. This last set of experiments could have been influenced by some water evaporation leading to an increase in the storage modulus as long as the gels did not undergo a gel–sol transition.

Pull-off experiments of PDA–gelatin gels between two stainless-steel plates were performed with 300 µL of hot sol being directly poured on the lower steel plate. The geometry was then closed with a constant gap of 0.5 mm between the lower and the upper steel plates. The gel was aged for 1 h at 25 °C before starting the peeling experiments. After this gelation time, measurements were undertaken by first compressing the gel under 1 N for 1 s. The upper geometry was then retracted with speeds of 1000, 100, and 10 µm/s at 25 °C, the force being recorded every ms until reaching a value of 0 corresponding to the rupture of either the adhesive bonds between the gel and one of the steel plates or the cohesive rupture of the gel itself. The type of rupture was determined via visual inspection (and digital photographs) after the total separation of the two plates. The tensile work per unit area was obtained by integrating the area under the force–separation curve and normalized according to the contact area between the hydrogel and the upper plate, a stainless-steel plate measuring 1 cm in radius (*A* = 3.14 cm^2^).

### 2.4. Thermal Treatment for Self-Healing

PDA–gelatin gels (20 mL) were formulated and aged for 1 h in a closed glass bottle. After rinsing, the gel-containing bottle was put on a heating plate at 80 °C for 30 min with 10 mL of MiliQ water under magnetic stirring at 200 rpm, achieving a final internal gel temperature of 55 °C (as recorded with a thermometer). Directly after heating, the self-healing of the gel was demonstrated by mixing pieces of gels together with a spatula after cooling the gel to room temperature. The same experiments were performed on unheated PDA–gelatin gels as well as on heated gelatin gels.

### 2.5. Antioxidant Activity Evaluation

The antioxidant activity of the PDA–gelatin gels was measured in accordance with [31], with only slight modifications. The gels were immersed in 30 mL of 2,2-diphenyl-1-picrylhydrazyl (DPPH, 10^−4^ M in absolute EtOH) (Sigma Aldrich, ref: D9132). For that aim, 1 or 2 mL of gel was put in contact with the DPPH solution, in the dark. The determination of the discoloration kinetics of the purple DPPH turning into a yellow product, translating into its reduction due to electron and proton transfer [32], was followed by UV-visible spectroscopy (SAFAS Monaco UVmc^2^) at λ = 516 nm, with absolute EtOH as the reference for the DPPH control, and the DPPH control as the reference for the samples. The volume of DPPH taken from the 30 mL in contact with the gels was put back in the reaction medium after each measurement to continue the determination of the kinetics. These experimental conditions were chosen, after qualitative tests, to ensure that DPPH was always in excess and could not be completely quenched and hence discolored.

To evaluate the PDA–gel antioxidant power over a prolonged duration of storage, several samples of independently made gels were formulated on the same day and stored in a humid and dark environment (but in the absence of excessive liquid) for 1, 2, 7, and 15 days. At those times, the DPPH conversion was measured via UV-visible spectrometry after 4 and 48 h hours of contact with the gel.

### 2.6. Statistical Analysis

Student’s *t*-test was performed to evaluate the statistical difference between the slopes of lnW versus lnV curves for gelatin and PDA–gelatin gels. One-way ANOVA tests were used to analyze the cohesive energy values of both kinds of gels, and the results are shown as mean ± one standard deviation (with **: *p* < 0.01; *: *p* < 0.05).

## 3. Results

### 3.1. Gel Formulation

The formulation of PDA–gelatin hydrogels is fast, simple, occurs in one pot, and does not require specific equipment. It only needs to be stirred and heated until the gelatin has dissolved at around 45 °C. An immediate color change from slightly beige to dark red and then to black (after a few minutes) occurs upon the addition of NaIO_4_ to reach a final NaIO_4_/DA ratio of 2 (DA stands for dopamine). This ratio was the chosen constant owing to the fact that dopamine oxidation entails two electrons as well as the two-electron reduction of periodate [28]. However, the oxidation of DA is not finished after its oxidation in dopamine quinone, hence more electrons are released and used to reduce more periodate anions. This requires a NaIO_4_/dopamine ratio larger than one, but a too-high value induces the oxidative degradation of the obtained PDA [28]. In addition, we previously found that the mechanical properties of gelatin@dopamine hydrogels, including their adhesion to steel, are optimal for a NaIO_4_/DA ratio close to 2 [21].

Visually, the result of gelation is a smooth shiny black stiff gel, that is easy to mold and to cut as needed. It can be dried via immersion in EtOH, in a 37 °C oven, or via freeze-drying, with a mass loss of 24-30-91%, respectively. Freeze-drying and EtOH treatment change the appearance of the gel by making the surface rougher in one case and deforming as well as stiffening it in the other. A gentler method such as drying in an oven keeps the shiny side of the gel while only thinning it. Simply immersing it in water makes it regain its original shape in all cases. Those dehydration methods can be used to store the material or make it easier to handle for further use, provided it keeps its adhesive properties.

### 3.2. Stiffness and Thermal Stability

A comparison between porcine and bovine gelatin in PDA–gelatin gels was performed by measuring their storage (G′) and loss modulus (G″). A PDA concentration of 1.5 mg/mL was chosen for the entirety of this article, as it is the lowest concentration where both pork and beef gelatin gels are stable at 37 °C and for higher temperatures, a requirement for future biological evaluations. The gelation kinetics exhibit significant differences between the two gelatins (Appendix A). For the one of porcine origin, the sol–gel transition happened before recording the data. Knowing that the time required to start data acquisition after deposition of the gel on the lower steel plate and closing the geometry with the upper plates takes about 90 s, the gelation time of porcine gelatin is hence shorter than this duration. However, for bovine gelatin, the gelation time is equal to 530 ± 141 s (n = 3) (Figure 1a).

After 1000 s of gelation, the G′ and G″ values differed by an order of magnitude between the porcine-based and bovine-based gelatin gels, the former being the stiffest (Figure 1a). This tendency for stiffer porcine-based gels (both containing PDA resulting from the NaIO_4_-triggered oxidation of DA) was also demonstrated during frequency sweep and thermal sweep experiments, where both gels were stable at low frequencies and at least up to 50 °C, the highest temperature measurable with the rheometer (Figure 1b,c). The differences between the porcine- and bovine-based gels are related to the 300 g bloom strength compared to the 225 g bloom strength, as well as their extraction method (Type A or B), respectively. For the following in this article, the porcine gelatin was selected for its lower t_sol-gel_, its higher stiffness, and more importantly its higher thermal stability (see Appendix A), which are better from a biomedical point of view as well as for other applications in which the materials can be subjected to temperature fluctuations between the ambient and 50 °C. It is apparent in Figure 1c that both gels do not undergo the gel–sol transition anymore in the presence of PDA, but that the storage modulus (G′) of porcine gelatin is almost not affected up to 50 °C. However, at higher temperatures (not controlled with the used rheometer) the porcine gel becomes less viscous but can still be turned upside down without flowing; hence, it remains a gel. From now on, the focus will be placed on porcine-based gels.

### 3.3. Strength of Adhesion to Stainless Steel

Pull-off tests were run on the PDA–gelatin gels, which were poured in between stainless-steel plates and left to gel for 1 h. The results were compared with those of the same gel without PDA in Figure 2A. By plotting the adhesion force of PDA–gelatin gels during peeling experiments, multiple breaks could be seen as *F* increased towards zero (Figure 2A), corresponding to the total rupture of all adhesive bonds. Such multiple-bond rupture phenomena do not occur for gelatin-only gels, where the force passes through a minimum, allowing a calculation of the adhesion strength, and then decreases continuously to zero. The tensile work per unit area, *W*, obtained via the area under the peeling curves and normalized according to the area of the upper plate, is increased by almost 2.5 and 4.5 times when PDA is blended into the gel at a 1000 µm/s and 10 µm/s peeling rate, respectively. The highest tensile work per unit area was 100 ± 39 J/m^2^ when the peeling speed was of 1000 µm/s for the PDA–gelatin gels. This value is higher by an order of magnitude than the value of 5.87 ± 0.45 J/m^2^ obtained by Wu et al. [22] for poly(HEMA-co-DMA) hydrogels containing 10.7% of side chains modified with catechols, the adhesion being measured on gold. Our results indicate that extremely strong adhesion can be obtained by blending the gelifying molecules (DA-oxidized in PDA) with the adhesive followed by oxidation instead of grafting the catechols on the polymer chain. However, our experiments were performed on steel and the data can hence not be directly compared with those of the adhesion tests performed with poly(HEMA-co-DMA) hydrogels on gold [22]. The highest tensile work per unit area obtained in the present investigation is, however, much lower than the 2400 J/m^2^ obtained for PDA–polyacrylamide hydrogels containing 8% of PDA in mass [33]. In this previous investigation [33], PDA was, however, synthesized before its incorporation in the acrylamide solution and its subsequent polymerization–gelation. In the present investigation, DA was added to the gelatin sol to reach a final concentration of 2 mg/mL; hence, this was only 0.2% (*w/w*). This small weight fraction may partially explain the lower *W* measured herein when compared to that of PDA–polyacrylamide hydrogels [33].

The tensile work per unit area for the unmodified gelatin gel also increased significantly with the peeling speed (Figure 2B) but always remained lower than that for the PDA–gelatin gels. In addition, in all cases, the rupture was cohesive (Appendix A) meaning that the adhesion with the steel plates was stronger than the gel structure. A cohesive rupture of PDA-containing hydrogels [20,21,33] or of hydrogels containing a grafted and oxidized catechol group [14,15] seems to be a general finding and highlights the high adhesion of those hydrogels to various solid supports [22], the weak point of the adhesive joint being the hydrogel itself. The tensile work per unit area may then be improved via the incorporation of fillers such as hydroxyapatite [15].

For purely elastic materials, *W* does not depend either on the contact time between two materials or on the peeling rate, at least in a very broad range of values. Such peeling rate dependencies are characteristic, among other mechanical properties, of viscoelastic materials; the tensile work per unit area between two cetyltrimethylamonium bromide coated monolayers on mica displays a 0.5 power law when ln*W* is plotted against ln*V* [23]. The same power law dependency has been found for polyhydroxyethylmethacrylate hydrogels modified with dopamine in the side chain at up to 10.7% (mol/mol) [22]. In the present investigation, the power law dependency is much less pronounced, being equal to 0.29 ± 0.03 and 0.16 ± 0.04 for pristine gelatin gels and for PDA–gelatin hydrogels, respectively. Those slopes are statistically different (*p* < 0.05) based on Student’s *t*-test. Interestingly, the peeling rate dependency is stronger for the unmodified gel, which may be explained by its higher G” value relative to the G’ value (quantified by the loss tangent value, Figure 1d) when compared to those of the PDA–gelatin gel. Both kinds of gels, even if they display a marked power law dependency of ln*W* versus ln*V*, nevertheless deviate significantly from the 0.5 power law dependency found by others [22,23]. We assume that this phenomenon originates from an elastic behavior that is predominant with respect to the viscous behavior responsible for energy dissipation in both gels, being more so when PDA is present as a “crosslinker” in the gel (Figure 1a,d).

### 3.4. Thermal Treatment for Self-Healing and Adhesion

Once poured, the PDA–gelatin gel is no longer as adhesive as is a freshly prepared gel. Indeed, its strength of adhesion to steel is 7.5 times lower than that of a “freshly” prepared gel in between steel, but almost 2 times higher compared to that of a simple PDA–gelatin gel membrane poured apart from steel. In agreement with this observation, when a PDA–gelatin gel is cut in pieces and if those pieces are reassembled, they do not stick firmly to each other. However, by submitting the PDA–gelatin gel to high temperatures (80 °C on the heating plate but 55 °C inside the bottle containing the gel), it becomes more liquid-like (without undergoing the gel–sol transition) and recovers some adhesive properties. We assume that, by lowering the viscosity of the gel, the heat allows the PDA molecules to move more freely within the mass of the gel and be more available on the surface to refresh the adhesion properties, at least partially. Directly after heating, it also gains self-healing properties that were not observed before heating (Figure 3a) and the gel can be remodeled as desired.

Gelatin gels without PDA simply re-gel after being heated at 80 °C and cooled at room temperature (Figure 3b). The newly obtained pieces of gelatin stick (Figure 3b) together, displaying the known self-healing behavior of gelatin [34]. This property is accorded to the PDA–gelatin hydrogels only after thermal treatment but without reaching the gel–sol transition), while PDA–gelatin hydrogels stored at ambient temperature do not display this effect.

The self-healing behavior of thermally treated PDA–gelatin gels but not of the unheated PDA–gelatin gels is of course only qualitative information but it suggests that the as-prepared gels can be stored and can recover a part of their adhesive properties by simply inducing some mobility (here emphasized by a decrease in G” upon heating, Figure 1d). We make the assumption that the thermal treatment of the PDA–gelatin gels allows a migration of part of the non-oxidized catechol groups present in the gel structure on their surface (Figure 2), allowing for the regeneration of adhesive properties. Such groups are present in the gel structure and contribute to its cohesion; indeed, during the peeling tests, the gels’ rupture was found to be cohesive. The validity of such an assumption will be tested in future investigations.

### 3.5. Antioxidant Activity

PDA, as seen previously, keeps its adhesive properties in a composite gel. The results showcased in Figure 4 demonstrate that this is also the case for its antioxidant activity, as found by O’Connor et al. [35] in polyethylene imine (PEI)–PDA gels crosslinked with dextran [33]. Indeed, after 4 h of contact between PDA gel and an ethanolic DPPH solution, the UV-visible kinetic spectra exhibit an 82% scavenging of DPPH radicals for 1 mL of gel and that of 83% for 2 mL of gel (Figure 4a), which is highly significant for the volume ratio considered, versus that of only 4% for gelatin gels without PDA (see Appendix A). The color change is noticeable to the naked eye, as seen in the picture in Figure 4. The difference between the two volumes of gel is not significant and can be attributed to the non-impregnation of DPPH within the gel, at least during the 2 h of the measurement. As a result, we assume that only the surface is in contact with the DPPH solution, and the values obtained were underestimated with respect to the scavenging effect that could have been obtained if the whole volume of the gel was reached by the bulky and pretty hydrophobic DPPH molecules.

However, by storing several samples of PDA gels for 1 to 15 days and measuring the DPPH conversion via UV-visible spectrometry after 4 and 48 h of contact, we can see that the gels keep an antioxidant effect with approximately 15% of conversion after 4 h of immersion in DPPH even after 15 days of storage, independently of the gel volume. After 48 h of permanent immersion in the DPPH solution, the values significantly increase to up to 40% for 1 mL of gel and to up to 60% for 2 mL of gel, which means that the PDA–gelatin gel retains important antioxidant activity, even after storage in an adequate environment, but with a slower kinetics compared to those of the initially prepared gels, as seen in Figure 4b. The same effect holds for gels which underwent the 80 °C thermal treatment described above. This phenomenon of the slow restauration of antioxidant properties can be of great interest in the biomedical field to control ROS during wound healing, for example. This also means that the gel can be stored and used or remodeled later without losing its properties. This slow antioxidant effect, coupled with the self-healing and adhesion results shown previously, demonstrates that molecules within the gel have some degree of freedom which can make them available given a proper environment and some prolonged time. Another possible explanation for the slow increase in DPPH scavenging for stored gels put in the presence of an ethanolic solution of DPPH is a slow release of gelatin (which would have no effect on the antioxidant activity) and PDA or PDA–gelatin conjugates (or clusters). To that aim, we performed a release experiment in which a PDA–gelatin gel was put in absolute ethanol for 48 h. The UV-vis spectrum of the supernatant above the gel, which undergoes shrinking in these conditions (see Section 2.1), is displayed in Appendix A. Some important absorbance between 200 and 230 nm was found but almost no detectable increase in absorbance above 250 nm was detected, implying that no dopamine (absorption peak at 280 nm) and no PDA (with a broad and peakless absorption spectrum in the UV-vis range [36]) are released in ethanol. The absorption spectrum displayed in Appendix A is consistent with the release of some gelatin but without a measurable release of PDA. Hence, the increasing DPPH scavenging upon long storage in its ethanolic solution cannot be attributed to a release of PDA particles or clusters in the supernatant but to a slow diffusion of DPPH in the gels, a diffusion that may be hindered with respect to the measurements made immediately after gel preparation due to a gel shrinking in ethanol.

## 4. Discussion

The research presented in this article shows that gelatin gels can be made much more adhesive to stainless steel by blending the gelatin sol with dopamine, which is oxidized and transformed into PDA via the addition of sodium periodate. The tensile work of adhesion is of the same order of magnitude as the values obtained for hydrogels where the catechol or dopamine group has been tethered to the polymer chains [8,9,10,11,12]. The rupture of the adhesive joint was of a cohesive nature, as is the case for other similar materials [14,15,20,21]. The dependency of the tensile work per unit area with the peeling rate is a power law with an exponent that is significantly lower than the value of 0.5 reported for viscoelastic gels [22] or surfactant-based monolayers [23] and is interpreted as the dominant elastic behavior of bovine gelatin gels (with a 300 bloom), which is even more the case when the hydrogel contains PDA. The viscoelastic behavior of the gelatin and PDA–gelatin gels is, however, not totally lost since both materials, and as expected for gelatin, are able to undergo self-healing when the initially prepared gel is heated and then cut into pieces before being re-assembled (Figure 3). The advantage of PDA–gelatin with respect to the gelatin gel is its improved thermal stability (Figure 1c) which will allow biological evaluations under the conditions of body temperature in the future. The self-healing and dynamic aspect of the PDA–gelatin hydrogels also accords prolonged antioxidant activity after the extended storage of the hydrogel. Indeed, when these hydrogels are stored under wet conditions, they are less antioxidant than are the freshly prepared ones even after only 2 days of storage when DPPH scavenging is measured for only 4 h, but more than 60% of the initial DPPH scavenging is recovered when the measurement is performed over 48 h (Figure 4b). This suggests that the antioxidant moieties in the gel become progressively accessible to DPPH (which is dissolved in ethanol). Nevertheless, this effect cannot be attributed to the release of PDA [31,35], in ethanol (Appendix A).

Furthermore, such preserved and prolonged antioxidant activity (in wet conditions but not in the fully hydrated state) is of the highest interest for biomedical applications where the ROS level after injury has to be controlled over a prolonged duration [25]. If such gels are used for protection against corrosion (however, using a controlled thickness for the coating) a reservoir of corrosion inhibitors that are slowly available but remaining in the coating could be of major interest.

## 5. Conclusions

The PDA–gelatin hydrogels prepared via the simple blending of a hot gelatin sol, dopamine and sodium periodate allow strong and peeling-rate-dependent adhesion between stainless-steel plates after only one hour of gelation. After storage under wet conditions, those hydrogels display self-healing properties only after thermal treatment, which has been attributed to a gel rearrangement during which groups responsible for gel cohesion diffuse to the surface and accord adhesion. As expected, the PDA–gelatin but not the pristine gelatin hydrogels are antioxidant owing to the presence of the PDA material. The most interesting finding is that this antioxidant activity is strongly preserved upon prolonged storage (15 days) but takes a long time (up to 48 h) to reach saturation owing to the slow diffusion of DPPH in the hydrogel and not to a slow PDA release in the DPPH solution.

## Data Availability

The experimental data displayed in this article can be made available upon contacting the corresponding author.

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
