# Peer review of "Strongly Metal-Adhesive and Self-Healing Gelatin@Polydopamine-Based Hydrogels with Long-Term Antioxidant Activity"

_antioxidants, 2023, doi:10.3390/antiox12091764_

Round 1

Reviewer 1 Report

Comments:

Editing of English grammar is needed. Some typos and errors are present in the manuscript

The authors should better highlight the novelty of the work by reporting a comparison with similar system already investigated and underlying the advantages of the proposed methodology

A simple scheme of the hydrogel preparation showing the crosslinked moieties can help understanding the adopted production process

The mechanism of self healing should be better explained. How does the chemical structure of PDA-Gel hydrogel affect it?

Sorption and/or water swelling data could be useful to further characterize the hydrogel since its potential application

Extensive editing of English language is required

Author Response

We would like to acknowledge reviewer 1 for his/her constructive comments about our manuscript. We reply in a point-by-point manner below. In the revised version of the article our additions corresponding to those requests appear in blue.

*Point 1: Editing of English grammar is needed. Some typos and errors are present in the manuscript.

Our reply: We totally agree and fixed the typos all along the revised version of the manuscript.

*Point 2: The authors should better highlight the novelty of the work by reporting a comparison with similar system already investigated and underlying the advantages of the proposed methodology.

Our reply: We totally agree and add a new sentence on lines 45-47: “Single blending of catechols or catecholamines with a polymer able to undergo gelation will all to produce strong adhesives in a one pot manner and to investigate the influence of the catecholamine structure and concentration on the hydrogel’s adhesive properties.

*Point 3: A simple scheme of the hydrogel preparation showing the crosslinked moieties can help understanding the adopted production process.

Our reply: This is an excellent idea and such a scheme has been added in the revised version of our article, in section 2.1 as well as a sentence on line 96 (page 2) of the revised version: “The whole experimental procedure is shown in Scheme 1.” The figure legend has been added on line 99.

 *Point 4: The mechanism of self-healing should be better explained. How does the chemical structure of PDA-Gel hydrogel affect it?

Our reply: This is an excellent suggestion made by the reviewer. To explain our hypothetical model for the self-healing mechanism we add a new scheme, which will be Scheme 2 in the revised version, page 8. We also add some text between lines 332 and 337 on pages 8-9 of the revised article: We make the assumption that a thermal treatment of the PDA-gelatin gels allows a migration of part of non-oxidized catechol groups present in the gel structure on their surface (Scheme 2) allowing for the regeneration of adhesive properties. Such groups are present in the gel structure and contribute to its cohesion, indeed during the peeling tests the gels’ rupture was found to be cohesive. The validity of such an assumption will be tested in future investigations.

*Point 5: Sorption and/or water swelling data could be useful to further characterize the hydrogel since its potential application.

Our reply: Basically we agree with the reviewer, but the gels produced in this investigation are not aimed to be used for bio-applications but for sealents of metallic pieces, they will hence not be put in the presence of water or aqueous solutions but eventually in the presence of a wet atmosphere. We add a sentence in the revised manuscript (lines 118-119, page 3): “No systematic swelling investigation was undertaken in this work because we aim to use these gels as adhesives for metallic materials and not for bio-applications.”

Reviewer 2 Report

 The aim of the present work is to improve the adhesion strength of gelatin hydrogels blended with dopamine and sodium periodate and to study the influence of the peeling rate on the adhesion strength. Authors showed that the PDA-gelatin gel is a promising, strong, and antioxidant adhesive. However, I have a few serious comments to this paper.

Line 70-74: In the last paragraph of introduction the aim of the manuscript should be presented not the results. This sentence should be removed from the introduction. The aim should be included here not in the line 45-48.

Figure 2 b. The linear regression from 3 points is not allowed. What was a coefficient of correlation. Is the coefficient was statistically significant?

Authors should also include the section conclusion.

In this manuscript authors have not included the statistical evolution of data. This disqualified this manuscript for publication.

To sum up, this manuscript should be rejected and the authors should be encourage to resubmit the corrected manuscript.

Author Response

We would like to acknowledge reviewer 2 for his/her constructive comments about our manuscript. We reply in a point-by-point manner below. In the revised version of the article our additions corresponding to those requests appear in red.

The aim of the present work is to improve the adhesion strength of gelatin hydrogels blended with dopamine and sodium periodate and to study the influence of the peeling rate on the adhesion strength. Authors showed that the PDA-gelatin gel is a promising, strong, and antioxidant adhesive. However, I have a few serious comments to this paper.

*Point 1: Line 70-74: In the last paragraph of introduction the aim of the manuscript should be presented not the results. This sentence should be removed from the introduction. The aim should be included here not in the line 45-48.

Our reply: We totally agree with the reviewer and we remove the sentence describing our results. We now add a new sentence between lines 75 and 79 (page 2): “It is the major aim of this investigation to test the assumption if the PDA-gelatin hydrogels prepared in a one pot manner are able to display long term antioxidant properties, in addition to strong adhesion on stainless steel”

*Point 2: Figure 2 b. The linear regression from 3 points is not allowed. What was a coefficient of correlation. Is the coefficient was statistically significant?

Our reply: We totally agree with the reviewer and performed peeling tests (3 experiments in each condition either gelatin alone or PDA-gelatin) at a peeling rate of 250 µm/s. Hence, our linear regressions were performed on 4 points (corresponding to 12 experiments). Note that we now plot lnW as a function of lnV (Figure 2b) and not any more W as function of V in a double logarithmic scale. The addition of a supplementary point on the data set hardly modifies the slope of the curves, which we use to discuss their viscoelastic character: they are now equal to 0.16 ± 0.04 (0.15 in the first version given without standard deviation) and to 0.29 ± 0.03 for the PDA-gelatin and gelatin gels respectively. We also discuss the statistical significance of the values of those slopes: “Those slopes are significantly different (P<0.05) based on a Student’s t-test” (line 293, page 7 of the revised manuscript). Note that the slopes obtained from the linear regressions as well as the corresponding correlation coefficients have been included in the inset of Figure 2b.

*Point 3: Authors should also include the section conclusion.

Our reply: we totally agree and add a short conclusion (this was also required by the Editor) in lines 422-432 of the revised article: The PDA-gelatin hydrogels prepared by simple blending of a hot gelatin sol, dopamine and sodium periodate allow strong and peeling rate dependent adhesion between stainless steel plates after only one hour of gelation. After storage in wet conditions those hydrogels display self-healing only after thermal treatment which has been attributed to a gel rearrangement during which groups responsible for the gel cohesion diffuse to the surface and afford adhesion. As expected, the PDA-gelatin but not the pristine gelatin hydrogels are antioxidant owing to the presence of the PDA material. The most interesting finding is that this antioxidant activity is strongly preserved upon prolonged storage (15 days) but takes long time (up to 48 h) to reach saturation owing to a slow diffusion of DPPH in the hydrogel and not to a slow PDA release in the DPPH solution.

*Point 4: In this manuscript authors have not included the statistical evolution of data. This disqualified this manuscript for publication.

Our reply: We totally agree with the reviewer, but the only time where a comparison between two data sets is required concerns the adhesiveness of the gelatin and PDA-gelatin gels (Figure 2b). Hence, we evaluated if the tensile work of PDA-gelatin gels is significantly higher than that of the unmodified gelatin gels. To that aim we performed a one-way ANOVA test and found a positive result with P<0.05 at low peeling (10 µm/s) and high peeling rates (1000 µm/s) and a positive result with better significancy (P<0.01) at peeling rates of 100 and 250 µm/s. The values are included in Figure 2b. In this figure, the limits of the 95% confidence intervals have been added as well as the corresponding sentence in the Figure legend (“whereas the dashed lines correspond to the limits of the 95 % confidence intervals ‘’). In addition, a new paragraph is added in the Materials and Methods section (page 4, lines 174-177):

“ 2.6 Statistical analysis

Student’s t-test was performed to evaluate the statistical difference of the slopes of lnW versus lnV curves for gelatin and PDA-gelatin gels. One-way ANOVA tests were used to analyze the cohesive energy values of both kinds of gels and the results are shown as mean ± one standard deviation (with : **: P<0.01 and *: P<0.05).”

Reviewer 3 Report

1.The link to the supporting material provided in the article " www.mdpi.com/xxx/s1 " suggests an error and that important material such as Figure S14: Antioxidant properties of gelatin gel compared to the PDA-gelatin gel should be inserted in the text.

2. Background descriptions for PDA can be strengthened by citing 10.1016/j.cej.2022.135691; 10.1016/j.carbpol.2021.118046 and what are the advantages of the current work compared to published articles?

3. The hydrogels synthesized in the article have long-term antioxidant activity properties, but they were characterized only with DPPH experiments, which is clearly insufficient and should be combined with data from another ABTS experiment.

4. The article mentions that freeze-drying and ethanol treatments can have an effect on the gel, so a milder alternative was chosen, such as drying it in the oven and immersing it in water to restore it to its original state. Should the temperature of the oven and the time spent in the oven be considered for the hydrogel? Whether or not the properties of a gel that has been oven-dried and restored to its original state have been affected cannot be concluded simply by looking at the appearance, but should be supplemented with some characterization data after the treatment.

5. The gel is highly adhesive to stainless steel and may have biomedical applications. Should the authors then consider its adhesion to special tissues of human skin authors and other effects?

Author Response

We would like to acknowledge reviewer 1 for his/her constructive comments about our manuscript. We reply in a point-by-point manner below. In the revised version of the article our additions corresponding to those requests appear in green.

*Point 1: The link to the supporting material provided in the article " www.mdpi.com/xxx/s1 " suggests an error and that important material such as “Figure S14: Antioxidant properties of gelatin gel compared to the PDA-gelatin gel” should be inserted in the text.

Our reply: we totally agree and the supporting material is now available with the revised version of the manuscript. We apologize for the non-availability of these data after the first submission.

* Point 2: Background descriptions for PDA can be strengthened by citing 10.1016/j.cej.2022.135691; 10.1016/j.carbpol.2021.118046 and what are the advantages of the current work compared to published articles?

 Our reply: We acknowledge the reviewer for the suggestion to add these two interesting references describing the application of polydopamine containing hydrogels for wound healing. They are now reference 29 and 30. All the subsequent reference numbers have been updated.

*Point 3: The hydrogels synthesized in the article have long-term antioxidant activity properties, but they were characterized only with DPPH experiments, which is clearly insufficient and should be combined with data from another ABTS experiment.

Our reply: Basically we agree, if we would quantify the anti-oxidant activity for instance in terms of “Trolox equivalents”. But herein we only want to display the prolonged antioxidant property upon storage of the PDA-gelatin hydrogels for days in a wet environment. This is a new finding, to our opinion. Hence, we believe that an evaluation based on the DPPH scavenging assay is significant.

 *Point 4. The article mentions that freeze-drying and ethanol treatments can have an effect on the gel, so a milder alternative was chosen, such as drying it in the oven and immersing it in water to restore it to its original state. Should the temperature of the oven and the time spent in the oven be considered for the hydrogel? Whether or not the properties of a gel that has been oven-dried and restored to its original state have been affected cannot be concluded simply by looking at the appearance, but should be supplemented with some characterization data after the treatment.

 Our reply: Indeed, we do not aim to show that the hydrogels can be restored in their original state (after oven drying and rehydration), we wanted to dry them in order to use them after prolonged storage. They restore part of their original antioxidant activity but not totally (Figure 4). This can be convenient for future users: they can prepare a large amount of such an hydrogel, dry it and use them as an adhesive (after heating, Figure 3) or antioxidant materials a few days later (Figure 4).

*Point 5. The gel is highly adhesive to stainless steel and may have biomedical applications. Should the authors then consider its adhesion to special tissues of human skin authors and other effects?

Our reply: We agree with the reviewer but before performing adhesion tests on human skin or other biological tissues; cytotoxicity assays need to be performed. In the title we restricted ourselves to “metal adhesive materials”.

Round 2

Reviewer 1 Report

The authors modifed the manuscript according to the proposed comments. I can now recommend its publication. 

Reviewer 2 Report

The authors corrected the manuscript accordingly.